# Three-Dimensional Analysis of Upper and Lower Arches Using Digital Technology: Measurement of the Index of Bolton and Correspondence between Arch Shapesand Orthodontic Arches

**DOI:** 10.3390/dj11080188

**Published:** 2023-08-08

**Authors:** Marco Pasini, Elisabetta Carli, Federico Giambastiani, Maria Rita Giuca, Domenico Tripodi

**Affiliations:** 1Unit of Pediatric Dentistry, Department of Surgical, Medical, Molecular and Critical Area Pathology, University of Pisa, 56126 Pisa, Italy; marco.pasini@med.unipi.it (M.P.); elisabettacarli1@gmail.com (E.C.); mariarita.giuca@med.unipi.it (M.R.G.); 2Department of Medical, Oral and Biotechnological Sciences, Dental School, University “G. D’Annunzio” of Chieti-Pescara, 66013 Chieti, Italy; tripodi@unich.it

**Keywords:** digital models, Bolton index, orthodontics CAD software, extraoral scanner

## Abstract

Introduction: Thanks to the great development of digital technology, viaCAD (computer-aided design) and CAM (computer-aided manufacturing) systems, digital models canbe used as an aid for orthodontic planning decision-making processes as there are numerous studies in the literature that support the validity ofthe digital model measurements of anterior teeth and the total coefficient of Bolton analysis. The aim of the present study isto compare the average length value of the current upper and lower arches with that of a hypothetical nickel–titanium wire and to confirm the reliability and accuracy of digitally taken measurements of the anterior and total Bolton coefficients.In this retrospective study, dental casts of 138 Caucasian adolescent patients were scanned with an extraoral scanner, and Ortho3Shape software was adopted for the following dental cast measurements: actual and ideal lengths of the lower arches and anterior and total Bolton coefficients.In the present study, we found that the mean value of the anterior coefficients of the Bolton index was compatible with those of previous studies, confirming the reliability of digital measurements.Therefore, digital CAD/CAM models may be a viable alternative to plaster models, as they can facilitate model preservation and recovery. For future studies, it would be better to use intraoral scanners (IOSs) to ensure greater accuracy, since they only require one step and allow obtaining better results for the patients.

## 1. Introduction

The development of digital technology, viaCAD (computer-aideddesign) and CAM (computer-aided manufacturing) systems, contributed to the improvement and simplification ofboth diagnosis and treatment planning in orthodontics.

The three-stage process of the digital workflow includes the following: the acquisition of digital images of patients’ dental arches, the manipulation of these images usingspecific dental software, and finally, the 3D printing files [1].

Orthodontic measurements from digital study models can be useful for assessing the need for orthodontic treatment and dental parameters, such as the Bolton index, and as already demonstrated in the literature, all analysed dental casts can potentially be transformed into 3D images [2].

The Bolton analysis is carried out by orthodontists to measure possible disproportions between the upper and lower dental arches and was developed by Wayne Bolton in 1958 to measure the ideal ratio between the mesiodistal diameters of the teeth of the upper arch compared to those of the lower arch. The mesiodistal diameters, at the contact points, of all elements from the central incisors to the first molars must first be calculated for each dental arch.

Digital models could be used in decisionmaking and treatment planning [3,4], as there are many studies in the literature that support the validity of the digital measurements of the dental arches for the measurements of both the anterior and total coefficients of the Bolton analysis [5,6].

As demonstrated in the scientific literature [7], the rotational and translational movements of teeth can be analysed and reproduced with great precision using digital configurations, and in more complex orthodontic treatments, a preliminary virtual plan has the potential to allow a significant reduction in errors with a greater probability of predicting the final orthodontic outcome.

Furthermore, digital technologies and artificial intelligence may allow greater opportunities for such planning, as they can be applied in the early stages of the clinical examination to develop a simultaneous virtual plan of all stages of treatment [8].

Recently, it has been shown that 3D printing can be a method used todigitally design and then produce customisable clear aligners with greater precision as an alternative to conventional orthodontic appliances while also offering greater fit and effectiveness [9].

In the present study, using an extraoral scanner and specific dental software, the average length values of the upper and lower jaws of adolescent patients were compared with those of a hypothetical nickel–titanium wire, and the accuracy of the measurements of the anterior and total Bolton index coefficients were calculated.

In the present retrospective study, dental cast models were already available; therefore, we decided to use an extraoral scanner instead of an intraoral scanner.

The aim of the study was to establish the average measurements of upper and lower arch lengths and the anterior and total Bolton coefficients of an Italian adolescent population. Moreover, the aim of this study was to evaluate the reliability of digital dental measurements.

## 2. Materials and Methods

In the present study, all operations were carried out by the same operator who analysed the dental measurements of 138 dental casts using the Ortho3Shape software (Figure 1). 

The parameters that were measured are as follows: current length of the dental casts, length of a hypothetical nickel–titanium orthodontic wire, and anterior and total Bolton coefficients.

### 2.1. Selection Criteria

The inclusion criteria were as follows:-Caucasian ethnicity;-Permanent dentition;-Class I and absence of severe malocclusions;-Absence of dental extractions or extensive tooth reconstructions;-Absence of previous maxillo-facial surgery;-Absence of previous removable or fixed orthodontic treatments;-Three-dimensional dental casts scanned usingOrtho3shape software.

The exclusion criteria were as follows:-Presence of deciduous teeth;-Teeth with severe rotation;-Dental agenesis;-Oligodontia.

### 2.2. Clinical Procedures

In the present study, the mesiodistal diameter of each tooth from the right canine to the left canine was measured to obtain the anterior Bolton index, both in the upper and in the lower dental arch.

Furthermore, in order to measure the total coefficient, the mesiodistal diameter of each tooth from the right molar to the left molar was measured.

The mean value of the length of a hypothetical nickel–titanium orthodontic wire of the upper and lower jaws was calculated on the basis of the dental cast measurements obtained usingthe Ortho3Shape software.

Then, the mean value was compared with the length of upper and lower dental arches.

Furthermore, to confirm the accuracy of the measurements of the anterior and total Bolton coefficients, our measurements were compared with those found in the study by A. Anand Kumar et al. [10].

### 2.3. Statistical Analysis

Sample size calculation was performed with a confidence level of 95%, a power of 80%, a population variance of 1000, and a hypothesized difference of 10 based on the Bolton analysis of a previous study [10].

Each measurement was evaluated twice by the same operator, with an interval of 2 weeks between the two evaluations. Intra-observer variability was calculated by using Cohen’s Kappa coefficient, and we found that the K coefficient was between 0.9 and 0.97.

The mean values of digital casts were measured, and Student’s *T* test was adopted with a level of significance set at *p* < 0.05.

The statistical analysis was carried outusing theSPSS (Statistical Package for Social Sciences, Chigago, IL, USA) 22.0 program.

The mean values of the anterior and total Bolton coefficients were compared with those obtained by A. Anand Kumar et al. in order to assess possible differences [10].

The study conducted by A. Anand Kumar et al. included a sample of 50 adults with similar inclusion criteria of our study:-Full permanent dentition from right first molar to left first molar in both the upper and lower arches;-Absence of previous or current orthodontic treatment;-Absence of severe dental crowding [10].

Statistically significant differences were measured between the two groups by using Student’s *t* test (*p* < 0.05).

## 3. Results

The results of the present study are reported in Table 1, Table 2, Table 3 and Table 4. Moreover, the mean values of the anterior and total Bolton indexes were compared with those calculated by A. Anand Kumar et al. [10] with respect to CBCT and dental casts (Table 5).

### 3.1. Digitaldental Casts Analysis

We found that the length of the upper arch was 92.13 mm, while the mean value of a hypothetical length of a nickel–titanium orthodontic wire of the upper arch was 103.79 (Table 1).

Moreover, it was observed that the length of the hypothetical nickel–titanium orthodontic wire of the upper arch was 11.66 mm longer than the one calculated usingdigital casts.

We observed that the lower arch measured on dental casts was 88.26 mm, while the mean value calculated for the hypothetical length of a nickel–titanium orthodontic wire was 94.71 mm (Table 2).

It was observedthat the mean value of the hypothetical length of a nickel-titanium orthodontic wire of the lower arch was 6.45 mm longer than the one measured in the lower arch.

### 3.2. Bolton Analysis Comparison between Our Study and A. Anand Kumar et al.’s Study [10]

*Anterior coefficient (Bolton)*: In the study by A. Anand Kumar et al., the mean value of the anterior Bolton coefficient of the 50 examined subjects was 0.76 for both CBCT and dental casts measurements while we obtained a value of 0.79 [10] in our study (Table 5).

*Total coefficient (Bolton):* In the study by A. Anand Kumar et al., the mean value of the total Bolton coefficient of the 50 examined subjects was 0.91 for both CBCT and dental casts measurements, while in our study, the value was 0.94 [10] (Table 5).

### 3.3. Comparisonof Anterior and Total Bolton Coefficients between the Two Studies

A significant difference was observed by the student *t*-test between the two studies both for the anterior coefficient (*p*: 0.0028; *p* < 0.05) and for the total coefficient (*p*: 0.0009; *p* < 0.05).

## 4. Discussion

In the present study, dental casts were digitized usinga 3D extraoral scanner to obtain STL (stereolithography:standard triangulation language) files viaCAD/CAM technology, which allows the surface data of the dental arches to be saved on the computer asthree-dimensional images [11,12,13].

STL files can allow clinicians to quickly obtain information for diagnostic purposes [14] and to digitally measure arch width, dental arches discrepancies, Bolton index discrepancies, overjet, and overbite. Moreover, they can be used for the simulation of teeth movements [15,16] and allow an efficient digital workflow with significant time savings for the orthodontist [17,18].

In addition, the importance of using a computer-aided design and manufacturing system in the indirect orthodontic bonding technique was previously demonstrated in the literature, and it guarantees high accuracy and precision [19].

Furthermore, digital dental casts may be displayed with symmetrical bases in order to evaluate possible dental arch asymmetries and increase teeth measurement accuracy [20].

In the present study, we evaluated the STL files of dental casts, and we performed a comparison between an ideal orthodontic wire and upper and lower dental arch lengths in a sample of adolescent patients.

We found that the average value of the length of a hypothetical orthodontic nickel–titanium wire of the ideal upper arch was 11.66 mm longer than the average value of the upper arch length that we measured on dental casts.

Furthermore, we observed that the average length value of anideal hypothetical lower arch nickel–titanium orthodontic wire was 6.45 mm longer than the average length value of the lower arch measured on dental casts.

A comparison was performed between the results of the present study and those found in the study of A. Anand Kumar et al. [10] that included 50 patients, and we observed that the mean values of both the anterior and total coefficient of the Bolton index were very similar [10]; therefore, we confirmed the reliability of the digital measurements performed usingorthodontic software. These measurements may be useful inestablishingthe average values of the upper and lower dental arches and the anterior and total Bolton index in an Italian adolescent population.

A limitation of the present study was the use of an extraoral scanner on wax dental casts instead of the use on an intraoral scanner. New intraoral scanners may be useful in future studiesthat aim toobtain dental arches measurements viaan easier and faster method.

## 5. Conclusions

The results of the present study showed that digital orthodontics and the reproduction of 3D digital models usingdental scanners are helpful for orthodontic diagnoses and treatment planning. These results confirm the findings of previous studies reported in the literature [17,18], and they are useful inestablishing the average measurements of the upper and lower arch lengths and the anterior and total Bolton coefficients of an Italian adolescent population.

In our study, the mean value of the anterior coefficients of the Bolton index was similar with those found in the study conducted by A. Anand Kumar et al., confirming the reliability of digital measurements [10].

However, in future studies, it will be preferable to use intraoral scanners (IOSs) that ensure greater accuracy, requiring only one step and ensuring a better result for the patient in comparison to extraoral scanners.

## Figures and Tables

**Figure 1 dentistry-11-00188-f001:**
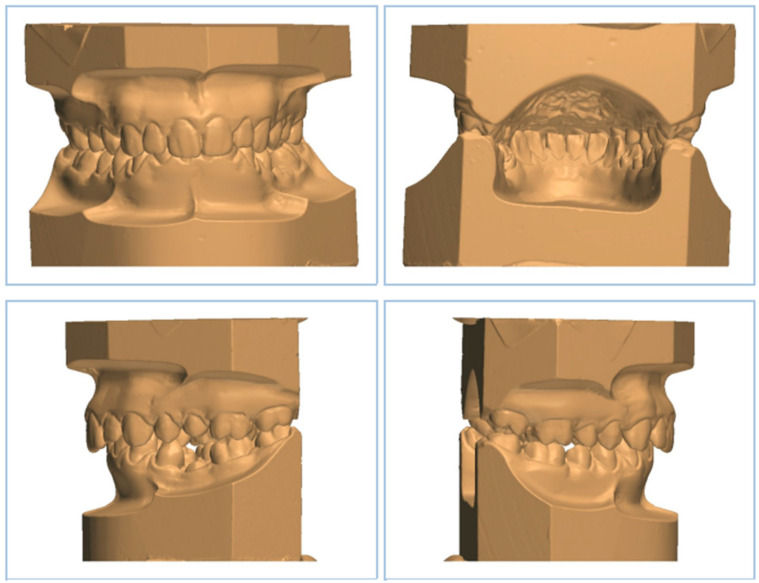
Three-dimensional digital models of dental casts.

**Table 1 dentistry-11-00188-t001:** Orthodontic wire and dental cast measurements in the upper arch.

Upper Arch Analysis	Hypothetical Nickel Titanium Orthodontic Wire Upper Archlength (mm)	Upper Arch Length (mm)
Maximum Value	124.33	108.07
Minimum Value	80.03	66.47
Mean Value	103.79	92.13

**Table 2 dentistry-11-00188-t002:** Orthodontic wire and dental cast measurements in the lower arch.

Lower Arch Analysis	Hypothetical Nickel Titanium Orthodontic Wire Lower Archlength(mm)	Lower Arch Length (mm)
Maximum Value	109.91	107.63
Minimum Value	75.32	68.47
Mean Value	94.71	88.26

**Table 3 dentistry-11-00188-t003:** Bolton Anterior Coefficient calculated in the present study.

Bolton Analysis: Anterior Coefficient	Maxillary Teeth (mm)	Mandibular Teeth (mm)	Bolton Value
Maximum Value	45.54	35.69	1.16
Minimum Value	24.33	22.58	0.62
Mean Value	38.16	30.10	0.79

**Table 4 dentistry-11-00188-t004:** Bolton Total Coefficient calculated in the present study.

Bolton Analysis: Total Coefficient	Maxillary Teeth (mm)	Mandibular Teeth (mm)	Bolton Value
Maximum Value	89.25	88.78	1.21
Minimum Value	58.10	53.39	0.68
Mean Value	78.99	73.77	0.94

**Table 5 dentistry-11-00188-t005:** Anterior and total Bolton coefficients calculated in the present study and in A. Anand Kumar et al.’study.

Mean Value	A. Anand Kumar et al.’s Study	Present Study	*p* Value
Anterior coefficient	0.76	0.79	0.0028
Total coefficient	0.91	0.94	0.0009

## Data Availability

The data that support the findings of this study are available from the corresponding author [Pasini, M.; Carli, E.; Giambastiani, F.; Giuca, M.R.; Tripodi, D.], upon reasonable request.

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
