# Peer review of "Three-Dimensional Analysis of Upper and Lower Arches Using Digital Technology: Measurement of the Index of Bolton and Correspondence between Arch Shapesand Orthodontic Arches"

_dentistry, 2023, doi:10.3390/dj11080188_

Round 1

Reviewer 1 Report

The present manuscript compared the average length value of the current upper and lower arches with that of a hypothetical nickel-titanium wire and the accuracy of the Bolton index measurement. The conclusions stated that CAD/CAM digital models can be a viable alternative to plaster models, considering the validity of digital measurements drawn by the present study.

Although the topic is not original, the study is well conducted and interesting. However, there are some points that need to be clarified:

-       I suggest improving the English to make the text more fluent and readable.

-       In the Introduction section, the authors should increase the bibliography.

-       In “Materials and Method” section, I suggest providing a better explanation of how the authors compared the arch length with a hypothetical nickel-titanium wire. I couldn’t understand it. 

-       Did the authors perform a sample size calculation?

-       I suggest reading a recent article (Fiorillo G, Campobasso A, Caldara G, Battista G, Lo Muzio E, Mandelli G, Ambrosi A, Gastaldi G. Accuracy of 3-dimensional-printed customized transfer tray using a flash-free adhesive system in digital indirect bonding: An in vivo study. Am J Orthod Dentofacial Orthop. 2023 Apr 18:S0889-5406(23)00167-1. doi: 10.1016/j.ajodo.2023.02.017. Epub ahead of print. PMID: 37074245.) about the accuracy of virtual planning of bracket positions (page 6, lines 208-217) to update the “Discussion” section.

It can be improved.

Reviewer 2 Report

Thank you for allowing me to review this retrospective laboratory study. Please see my suggestions below:

Abstract:

1.     Second paragraph of the abstract can be eliminated. Authors must remember that the abstract is the summary of the paper, and one or two sentences should suffice to justify the paper. 

2.     The aim of the study is vague; thus, I am curious to find out more about the aim while reading the manuscript

3.     The abstract does not clearly point it out the outcome of the study. I also lack a conclusion.

Introduction:

1.     Second paragraph, there is a repetition of the words;” patients' dental arches”.

2.     The introduction if truncated, and not clear.

3.     Paragraph 5, it states in the beginning of the paragraph: “as demonstrated in the literature” … however the authors don’t cite anyone. Please cite where the authors found the information.

4.     The aim of this study is interesting, however, with so many new intra-oral scanning technologies available, it is questionable why the authors decided to scan the dental casts with an extra-oral Ortho3Shape.

Subject and Methods

1.     If the authors are attempting to study the BOLTON discrepancy as a study goal, the exclusion criteria “Abnormalities of eruption or formation of dental elements” should not be used. If indeed the criteria was used, then determining if a Bolton discrepancy exist in any case becomes irrelevant. What the authors are attempting to investigate? Did they select 138 dental casts with normal occlusion, not anomalies, and then check if the Bolton discrepancy matches the Bolton results? Wayne Bolton collected a large sample of dental casts and measure them all to create a standard ratio measurement for maxillary and mandibular arches. And my guess is that the authors of this study are attempting to use these dental casts to measure a similar thing, but this time they used a standard NITI wire.

2.     The first sentence of the section 2.2 can be eliminated since it is stated in the first paragraph of the M&M.

3.     The authors should attempt to prevent from stating the same thing twice about the Bolton analysis. Please make it more objective and state on the paper only once, either when you’re describing your study or when you are presenting your parameters.

4.     The authors did not explain in any part of the materials and methods where and how the hypothetical Nickel Titanium wire measurement was included. And this is essential to the study, since it is part of the aim of the study. I could not find a description of the influence on the hypothetical NiTi orthodontic wire and the current upper arch length.

Results

1.     PLEASE ELIMINATE TABLE 6. No individual would like to see this table in the paper, this is why we have statistical analysis to calculate the overall study.

Discussion

1.     The following statements do not belong to this study since they have absolutely no relevance to the study: FROM line 203 to 222 can be eliminated since it is not pertinent to the study.

Conclusion - fair.

References

1.     Enough.

We suggest authors to find a fluent English speaking person to revise the grammar.

Reviewer 3 Report

Dear Authors, thank you for the recent study. I do believe that the study is interesting for the orthodontists and even the GP who use the CAD/CAM technology in virtual planning or analysis.

Language editing is recommended.

Round 2

Reviewer 1 Report

I would like to thank the authors for their revisions. Now, I think that this article may improve our knowledge in this topic.

Author Response

Plase see the attachment.

Reviewer 2 Report

Thank you for allowing me to review this retrospective laboratory study again. Please see my suggestions below:

Abstract:

1.     Abstract is now fine.

Introduction:

1.     Authors have improved.

Subject and Methods

All tables should have a description of what they represent. As I received the manuscript, the tables are only labeled with numbers. Furthermore, it would be nice if they were shown only at the end of the paper. However, I am not sure if this is a request from the journal.

Discussion is fine.

References:

Check this reference when talking about “Orthodontic measurements of digital study models may be useful in assessing orthodontic treatment need and dental parameters i.e. the Bolton index. “Bosio JA, Rozhitsky F, Jiang SS, Conte M, Mukherjee P, Cangialosi TJ: Comparison of scanning times for different dental cast materials using an intraoral scanner, J World Fed Orthod; 6 (2017) 11e14.

This reference might be useful to describe how digital dental casts may be used in the future.

Please attempt to show this manuscript to a native English speaker. It might help to improve the quality of your manuscript. Thank you.
